# Combination Therapy Is Not Associated with Decreased Mortality in Infectious Endocarditis: A Systematic Review and Meta-Analysis

**DOI:** 10.3390/microorganisms12112226

**Published:** 2024-11-02

**Authors:** Parisa Farahani, Felicia Ruffin, Mohammad Taherahmadi, Maren Webster, Rachel E. Korn, Sarah Cantrell, Lana Wahid, Vance G. Fowler, Joshua T. Thaden

**Affiliations:** 1Division of Infectious Diseases, Duke University, Durham, NC 27710, USA; pfarahani@carilionclinic.org (P.F.); rachel.korn@duke.edu (R.E.K.);; 2Research and Development, Carilion Clinic, Roanoke, VA 24014, USA; 3School of Medicine, Tehran University of Medical Sciences, Tehran 1461884513, Iran; taherahmadi.m@gmail.com; 4Medical Center Library & Archives, Duke University, Durham, NC 27710, USA; 5Department of Medicine, Carilion Clinic, Roanoke, VA 24018, USA; 6Department of Medicine, Virginia Tech Carilion School of Medicine, Roanoke, VA 24016, USA; 7Duke Clinical Research Institute, Duke University, Durham, NC 27710, USA

**Keywords:** combination therapy, endocarditis

## Abstract

Untreated infective endocarditis (IE) is uniformly fatal. The practice of combination antibiotic therapy for IE is recommended by treatment guidelines but largely unsupported by high-quality evidence. This study aimed to assess the efficacy of combination antibiotic therapy compared to monotherapy in IE through a systematic review and meta-analysis. We systematically searched MEDLINE, Embase, Cochrane, Web of Science, and CINAHL from inception to 29 July 2024. Studies reporting mortality outcomes of combination therapy versus monotherapy in adult patients with IE were included. Non-English papers and studies with less than 10 patients in the combination therapy group were excluded. Two reviewers independently assessed the studies and extracted relevant data. Summaries of odds ratios (ORs) with 95% confidence intervals (CIs) were evaluated using random-effects models. Out of 4545 studies identified, 32 studies (involving 2761 patients) met the inclusion criteria for the meta-analysis. There was no significant difference in the risk of all-cause mortality between the monotherapy and combination therapy groups (OR = 0.90; 95% CI = 0.67–1.20). Similar results were observed in subgroup analyses based on mortality time points, bacterial species, publication date, and type of study. Studies conducted in Europe reported a statistically significant decrease in overall mortality risk with combination therapy (OR = 0.67; 95% CI = 0.51–0.89), though this result was driven entirely by a single outlier study. Combination antibiotic therapy in patients with IE was not associated with reduced mortality.

## 1. Introduction

First described in Osler’s Gulstonian lectures, infective endocarditis (IE) remains a life-threatening condition [1,2]. One of the ongoing controversies in managing IE is the role of combination antibiotic therapy (more than one antibiotic therapy). In vitro studies have demonstrated the synergistic bactericidal activity of some antibiotic combinations, particularly for Gram-negative bacteria [3,4] and streptococci [5,6]. Animal studies have also demonstrated that combination therapy can achieve faster cardiac vegetation sterilization [5,6]. However, combination therapy may be associated with an increased incidence of adverse events [7,8]. To better understand the potential impact of combination antibiotic therapy, we performed a systematic review and meta-analysis of all studies that compared mortality between monotherapy (single-antibiotic therapy) and combination therapy in patients with bacterial IE. To identify potential subgroups of patients that may particularly benefit from combination therapy, we included studies involving any bacterial species. Prior systematic reviews addressing combination therapy in patients with IE have focused on only single bacterial species and specific antibiotic combinations [9,10].

## 2. Methods

This review focused on the following key question: *In patients with infective endocarditis, is combination antibiotic therapy associated with lower mortality than monotherapy?* We followed a standard protocol, developed prior to the literature review, for all steps of this review. The study protocol was registered on Prospero (CRD42023446243). The reporting of this systematic review was guided by the standards of the Preferred Reporting Items for Systematic Review and Meta-Analysis (PRISMA) 2020 statement [11].

### 2.1. Search Strategy

The MEDLINE (via Ovid, 1946–present), Embase (via Elsevier, 1947–present), Cochrane Central Register of Controlled Trials (via Wiley, 1998–present), Web of Science (via Clarivate, SCI-Expanded 1900–present, SSCI 1900–present, and ESCI 2019–present), and CINAHL Complete (via EBSCO, 1937–present) databases were searched from database inception to 28 July 2023. The search from database inception was performed to be inclusive and fully understand the available data evaluating monotherapy and combination therapy in IE. An updated search was performed on 29 July 2024, to identify more recently published manuscripts. An experienced medical librarian (SC) devised and conducted the searches, with input on keywords from the other authors. The search utilized a combination of database-specific controlled vocabulary terms and keywords searched in the titles or abstracts for the following concepts: endocarditis, antibiotics, and combination therapy. Case reports, comments, editorials, and conference abstracts were excluded. No other limits or restrictions were placed on the search. The search strategy was validated against a set of pre-selected articles and were independently peer-reviewed by a librarian using a modified PRESS Checklist [12]. The full, reproducible search strategies for all included databases are in Appendix A. All citations were imported into Covidence, a systematic review screening software [13].

### 2.2. Study Selection, Data Extraction, and Quality Assessment

We included only full-length published papers involving adults aged 18 years and older with IE in our analysis. Studies were eligible if they reported mortality outcomes and provided adequate data on monotherapy and combination therapy to enable the calculation of the odds ratio (OR) between treatment groups. Articles were excluded if they did not report outcomes, if outcome data were not extractable for each treatment group, if the combination therapy group had fewer than 10 patients, or if the article was not available in English. Two reviewers (from the following list: P.F., F.R., R.E.K., M.W.) independently screened references by title and abstract in the Covidence systematic review screening software. Included articles were independently screened by two reviewers (from the following list: P.F., M.T., F.R., R.E.K., M.W.) at the full-text level. Conflicts at both stages were resolved through discussion (P.F., F.R.). Data extractions were performed by two reviewers (from the following list: P.F., M.T., R.E.K., M.W.) independently and conflicts were resolved through discussion (P.F., M.T.). Extracted variables included lead author, journal, year of publication, start and end year of inclusion, country, study design, number of hospitals, number of patients, population description, inclusion criteria. For each treatment group within the studies, we also gathered data on the targeted organism(s), names and dosages of antibiotics used, affected heart valves, number of surgeries performed, relapse, treatment failure, and mortality rates at different reported time points. Mortality data were collected as the total number of patient deaths in each group.

Quality assessment and risk of bias were conducted by P.F. and verified by M.T. For randomized controlled trials, we used the Risk of Bias 2 (ROB 2) tool, which evaluates five domains: bias arising from the randomization process, bias due to deviations from intended interventions, bias due to missing outcome data, bias in the measurement of the outcome, and bias in the selection of the reported result [14]. For observational studies, we used the Newcastle–Ottawa Scale (NOS), which assesses the quality of nonrandomized studies based on three domains: selection, comparability, and outcome [15].

### 2.3. Data Synthesis and Analysis

Unadjusted mortality data were combined as odds ratios using Mantel–Haenszel with random effects models. We used the Knapp and Hartung method to adjust the standard errors of the estimated coefficients [16,17]. The robustness of the findings was assessed through influence and sensitivity analyses, as detailed in the text. We evaluated statistical heterogeneity with Cochran’s Q and *I*^2^ statistics. To explore potential sources of heterogeneity, we performed meta-analyses on subsets of studies to determine if variation in factors such as mortality time point (e.g., inpatient, 30-day, or 1-year mortality), bacterial groups (e.g., Gram-positive only or Gram-negative only), or geographic location between studies could be contributing factors. Statistical analyses were performed with RSstudio 2023.03.0. Publication bias was assessed using funnel plots with Egger’s test [18] when ≥10 studies were included in the analysis. We used the GRADE (Grading of Recommendations, Assessment, Development, and Evaluations) approach to evaluate the overall strength of evidence [19,20].

## 3. Results

### 3.1. Summary of Included Studies

The article selection flow diagram is presented in Figure 1. The searches yielded a total of 4545 citations after the removal of duplicates. Of these, 4195 papers were removed after a review of the title and abstract due to non-relevant disease states (i.e., not IE), patient populations (i.e., pediatric), or study types (i.e., in vitro studies, in vivo animal studies, case reports, review articles, or commentaries). Full-text assessment of the remaining 350 papers led to the exclusion of 319 papers. In total, we included 32 studies (2761 patients) in the analysis [21,22,23,24,25,26,27,28,29,30,31,32,33,34,35,36,37,38,39,40,41,42,43,44,45,46,47,48,49,50,51,52]. Summary data from the included studies and patients are shown in Table 1. Detailed characteristics of each study are available in Appendix A. There were 5 randomized controlled trials and 27 observational studies. Nearly half of the studies (44%, n = 14) were published before 2000. Aminoglycosides were employed as part of combination therapy in 11 studies, all of which targeted Gram-positive bacteria (Table 1).

Out of the 27 included observational studies, 10 were assessed to be of low risk of bias, while the remaining 17 were deemed to be at high risk of bias, mainly due to the lack of statistical adjustment between the monotherapy and combination therapy groups (Appendix A). Out of five randomized control trials, three had a high risk of bias and two had a low risk of bias (Appendix A).

### 3.2. Overall Mortality

There was no significant difference in the risk of all-cause mortality between combination antibiotic therapy and single-antibiotic therapy (OR = 0.90; 95% CI = 0.67–1.20). Low heterogeneity was observed in this analysis (π^2^ = 0.0829; *p* = 0.30; *I*^2^ = 11%) (Figure 2). To determine if combination therapy could be associated with mortality in particular groups, we performed meta-analyses on multiple patient subgroups. A funnel plot did not reveal evidence of significant publication bias (*p* = 0.11) (Appendix A).

### 3.3. Mortality by Different Time Points

No significant difference between combination therapy and monotherapy was observed when analysis was restricted to in-hospital mortality (n = 18 studies), 30-day mortality (n = 11 studies), or 1-year mortality (n = 7 studies) (Appendix A, respectively). Low heterogeneity was observed for these analyses.

### 3.4. Mortality by Different Bacterial Species

There was no significant difference in mortality between combination therapy and monotherapy for infections caused by Gram-positive bacteria (n = 23 studies) or Gram-negative bacteria (n = 8 studies) (Appendix A). Low and moderate heterogeneity were observed for these analyses, respectively.

### 3.5. Mortality by Location

A sub-analysis of 14 studies conducted in Europe demonstrated a statistically significant reduction in overall mortality risk with combination therapy (OR = 0.67; 95% CI = 0.51–0.89). Low heterogeneity was observed (π^2^ = 0; *p* = 0.52; *I*^2^ = 0%) (Figure 3). To determine if this result was due to an outlier study, an influence analysis was performed and showed that omission of the study from Escrihuela-Vidal et al. resulted in a non-significant pooled estimate (*p* = 0.19) [27] (Appendix A). There was no significant difference in mortality between combination therapy and monotherapy in studies conducted in United States (Appendix A). Low heterogeneity was observed for this analysis.

### 3.6. Mortality by Study Type

A subgroup analysis of observational studies did not reveal a difference in mortality between combination therapy and monotherapy (OR = 0.86; 95% CI = 0.64–1.15) (Appendix A). Similarly, no difference was observed when the analysis was restricted to randomized controlled trials (OR = 1.73; 95% CI = 0.52–5.79) (Appendix A). Low heterogeneity was observed for both analyses.

### 3.7. Mortality by Study Publication Year

To account for differences in antibiotic therapy regimens across time, subgroup analyses of studies published before and after 2000 were performed. In studies published before 2000, there was no difference in mortality between combination therapy and monotherapy (OR = 0.71; 95% CI = 0.40–1.25) (Appendix A). In studies published after 2000, there was similarly no difference in mortality between combination therapy and monotherapy (OR = 1.02, 95% CI = 0.70–1.47) (Appendix A).

### 3.8. Overall Evaluation of the Evidence

The evidence profile shows the overall strength of evidence from the meta-analysis of all included studies (Figure 4). Given that this systematic review contained primarily observational studies, the baseline strength of evidence was low. The certainty of evidence was downgraded due to risk of bias, as the included studies were primarily observational in nature and did not account for confounding variables. The certainty of evidence was upgraded as plausibly confounding variables (i.e., sicker patients receiving combination therapy as opposed to monotherapy) would reduce the demonstrated effect of combination therapy on mortality. The mortality effect estimate was not downrated due to inconsistency, indirectness, imprecision, or publication bias. We did not have serious concerns for inconsistency or imprecision as studies had generally similar odds ratios and the confidence interval of the pooled estimate was narrow. We did not have concern for indirectness as the evidence directly answered the question that was asked. We did not detect publication bias. Therefore, the overall strength of evidence for the association of combination therapy with mortality in patients with IE was low.

## 4. Discussion

While numerous systematic reviews and meta-analyses have examined combination therapy in bacteremia, such studies in IE have been limited to single bacterial species and specific antibiotic combinations [9,10]. This systematic review and meta-analysis involving all bacterial species and antibiotic combinations included over 2700 patients with IE and found no significant difference in mortality between combination antibiotic therapy and single antibiotic therapy. Similar results were observed when the analysis was restricted based on time points, bacterial species, and type of study. However, stratification by geographic region revealed decreased mortality with combination therapy in studies conducted in Europe.

Overall, we did not identify an association between combination therapy and decreased mortality in patients with IE. We believe that this study provides sufficient power to make such a determination. Basing effect size on the largest study included in this meta-analysis (n = 899 patients; 16.0% mortality with combination therapy; 21.8% mortality with monotherapy) [27], the overall combination therapy to monotherapy patient ratio of 1.66, and alpha 0.05, we would have >95% power to detect a difference between patients receiving combination therapy versus monotherapy with our overall study size. This result is similar to that of a prior meta-analysis that examined the role of adjunctive rifampin and gentamicin in patients with Staphylococcal prosthetic valve IE [9] and another that examined the role of adjunctive aminoglycosides with beta-lactam antibiotics [10]. In settings where a microbiological diagnosis has been made, potential benefits of combination therapy include synergy and the prevention of treatment-emergent resistance [53]. In synergy, antibiotic combinations are used for their synergistic action as the combined effects of two agents together are greater than the sum of their individual activities. For example, animal studies have demonstrated that combination therapy can achieve faster cardiac vegetation sterilization [5,6]. In treatment-emergent resistance, non-susceptibility to the antibiotic being used emerges in the setting of therapy. The bar for treatment-emergent resistance is presumed to be higher in combination therapy as opposed to monotherapy. However, the theoretical benefits of combination therapy in IE have not yet clearly been demonstrated to have an impact on patients with IE.

We found that combination therapy was associated with decreased mortality in studies conducted in Europe. We interpret this result cautiously as it was driven entirely by a single study by Escrihuela-Vidal et al. [27]. This retrospective cohort of 899 patients compared concomitant use of a beta-lactam and an aminoglycoside to monotherapy for the treatment of *viridans* and *gallolyticus group streptococci*. Overall, the study suffered from important limitations that limited the generalizability of the findings, including the fact that the combination therapy recipients were healthier overall, with a significantly lower mean age (64 vs. 72 years; *p* < 0.01) and burden of medical comorbidities (Charlson comorbidity index 1 vs. 2; *p* < 0.01). Perhaps most important, combination therapy recipients were significantly more likely to receive cardiac surgery (47% vs. 37%; *p* = 0.002), an intervention shown to improve survival [54]. And while the 1-year mortality in this cohort was lower with combination therapy relative to monotherapy, in-hospital mortality did not differ between the two groups. After omitting this study, an influence analysis revealed that the association between combination therapy and improved survival among the European studies disappeared.

This study had several limitations. First, while our study design allowed us to broadly address the impact of combination therapy on mortality in patients with IE, we may not have been able to detect individual antibiotic combinations that may have been beneficial in particular settings. For example, the role of combination therapy in prosthetic valve IE could not be determined, as studies typically involved mixed populations of native and prosthetic valve IE. Second, given the observational nature of most included studies, important confounding variables that impacted both the decision to employ monotherapy versus combination therapy and the patient outcomes may not have been accounted for. For example, the role of surgical intervention in managing such patients was not generally addressed in the included studies or by our meta-analysis. Another confounder was the type of valve as we were unable to provide a subgroup analysis in patients with prosthetic valve IE. Given the increasing prevalence of prosthetic valve IE in recent years [55], and the fact that many studies in this meta-analysis were conducted before 2000, the findings may not be fully generalizable to current cases of IE. Other confounders such as patient age, medical comorbidities, and others were similarly not generally accounted for in the observational studies. Finally, adverse complications of antibiotic therapy were not addressed in this study, though would be expected to be more common in those on combination therapy.

Ideally, future studies addressing the safety and efficacy of combination therapy in managing patients with IE will be randomized and controlled to better minimize bias between the treatment arms. Observational studies addressing the impact of combination therapy should account for confounding effects such as selection bias, treatment bias, and immortal time bias. Selection and treatment bias (i.e., surgical intervention) between the study arms should be accounted for with carefully chosen inclusion/exclusion criteria and statistical adjustment for confounding variables (e.g., propensity score-based approaches). Immortal time bias can be accounted for using Cox proportional hazards models with time-dependent treatment variables.

## 5. Conclusions

Combination therapy compared with monotherapy was not associated with decreased mortality in patients with IE. However, this systematic review and meta-analysis was limited by a high number of observational studies and a lack of stratification by factors such as surgical intervention and the type of valve involved in the infection.

## Figures and Tables

**Figure 1 microorganisms-12-02226-f001:**
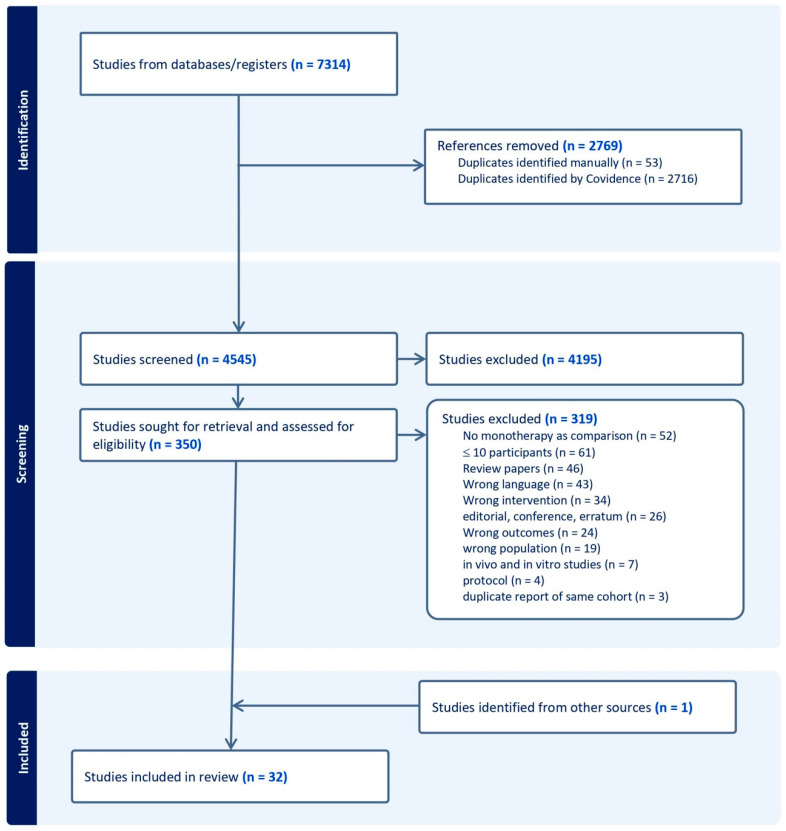
Preferred Reporting Items for Systematic Reviews and Meta-Analyses (PRISMA) flow diagram.

**Figure 2 microorganisms-12-02226-f002:**
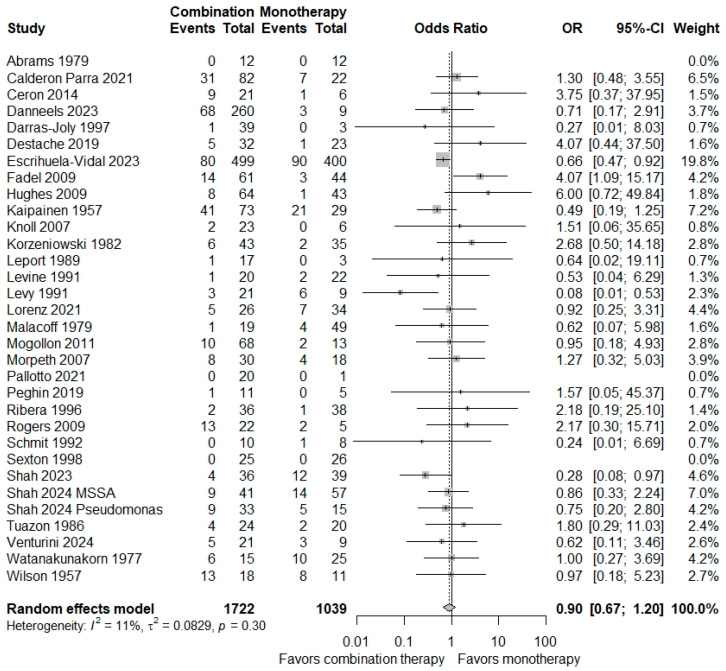
Forest plot of overall mortality in patients with infective endocarditis treated with monotherapy versus combination therapy. All included studies are shown here. The primary mortality endpoints (e.g., in-hospital mortality, 30-day mortality, etc.) are represented here [21,22,23,24,25,26,27,28,29,30,31,32,33,34,35,36,37,38,39,40,41,42,43,44,45,46,47,48,49,50,51,52].

**Figure 3 microorganisms-12-02226-f003:**
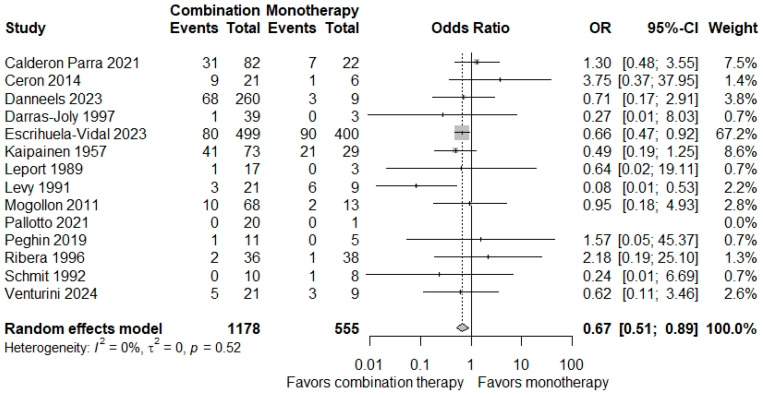
Forest plot of overall mortality in patients with infective endocarditis treated with monotherapy versus combination therapy for studies conducted in Europe. The primary mortality endpoint (e.g., in-hospital mortality, 30-day mortality, etc.) for each study is represented here [22,23,24,25,27,30,33,35,38,40,41,42,44,49].

**Figure 4 microorganisms-12-02226-f004:**
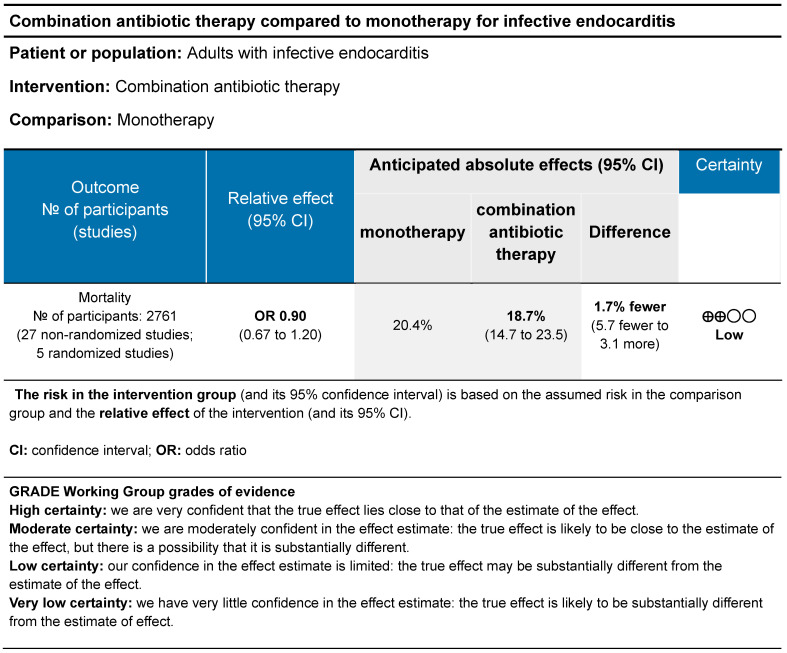
Evidence profile for impact of combination therapy on mortality in patients with infective endocarditis.

**Table 1 microorganisms-12-02226-t001:** Description of studies included in the systematic review.

Study Characteristics	Studies, No. (%) (N = 32)
**Publication year**	
1957–1999	14 (44)
2000–2012	6 (20)
2013–2024	12 (36)
**Study design**	
Observational Studies	27 (84)
Randomized controlled trials	5 (16)
**Country**	
United States	16 (50)
Europe	14 (44)
Australia	1 (3)
Multiple sites	1 (3)
**Number of patients**	
Monotherapy (median [range])	1039 (19 [1–400])
Combination therapy (median [range])	1722 (25.5 [10–499])
**Number of sites**	
1	10 (31)
2–20	11 (34)
>20	5 (16)
Unknown	6 (19)
**Bacterial species**	
Gram-positive only	
*Staphylococcus aureus*	10 (31)
*Streptococcus species*	6 (19)
*Enterococcus species*	4 (12)
Other Gram-positive	3 (9)
Gram-negative only	
*Coxiella Brunetti*	2 (6)
*Serratia marcescens*	1 (3)
*Pseudomonas aeruginosa*	1 (3)
Other Gram-negative	4 (12)
**Antibiotic therapy**	
Monotherapy	
Beta-lactam	14 (44)
Daptomycin	3 (9)
Doxycycline	2 (6)
Teicoplanin	2 (6)
Other	11 (34)
Combination therapy	
Beta-lactam + Aminoglycoside	8 (25)
Beta-lactam + other	4 (12)
Daptomycin + other	2 (6)
Teicoplanin + other	2 (6)
Other	16 (50)
**Timing of primary mortality outcome**	
In-hospital mortality	18 (56)
28–30 day mortality	11 (34)
1 year mortality	7 (22)
**Risk of bias**	
Low	12 (38)
Medium	0
High	20 (62)

## Data Availability

The data that support the findings of this study are available from the corresponding author, J.T.T., upon reasonable request.

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
