# Peer review of "Combination Therapy Is Not Associated with Decreased Mortality in Infectious Endocarditis: A Systematic Review and Meta-Analysis"

_microorganisms, 2024, doi:10.3390/microorganisms12112226_

Round 1

Reviewer 1 Report

Comments and Suggestions for Authors

Dear authors,

I greatly appreciate your great effort invested in this systematic review and meta-analysis. The manuscript is worthy and may significantly improve our approach to the management of bacterial endocarditis. My comments are the following:

1.  The title of the manuscript: I suggest adding "is" after "therapy", and to use capital letters in the beginning of each word in accordance with the style of the Journal.

2.  Please, rearrange the numbers of the references within the text in accordance with the style of the Journal {[1-4] instead of (!-4)}.

3.  Please remove "(single antibiotic therapy)" from line 75 to line 44 after the first appearance of "monotherapy", and "(more than one antibiotic therapy)" from line 76 to line 37 after the first appearance of "combination antibiotic therapy".

4.  References section: Please rearrange the references including the number of the authors in accordance with the style of the Journal.

I wish you good luck and much success.     

Reviewer 2 Report

Comments and Suggestions for Authors

Congratulations, huge work!

The results are rather well presented, but the relevance of the raised question is not current.

If we take into account the severity of IE and all the published guidelines, the probability that a clinician will not consider the guidelines and national and/or local protocols by using monotherapy is very low, except for randomised clinical trials. According to ESC 2023, only native valve infective endocarditis and some rare aetiologies, such as Coxiella burnetii, Mycoplasma pneumoniae, Legionella, and Tropherima whipplei, are to be treated with monotherapy.

All the RCT were performed before 2000; therefore, the inference for present days is less relevant, mainly because of the high frequency of prosthetic valve IE that are not treatable with monotherapy.

In Table 1 you mentioned Doxycycline and Daptomycin as monotherapy. Doxycycline is used only in Coxiella burnetii IE, and daptomycin cannot be used in monotherapy, just in combination with beta-lactams or fosfomycin. because of resistance issues.

The discussion section is rather short considering the complexity of the subject, and the limitations should be extended to other issues.

The conclusion is rather straightforward, and the results are not supporting it.

Reviewer 3 Report

Comments and Suggestions for Authors

The study refers to a meta-analysis of combined antibiotic therapy in endocarditis. Although the manuscript is well-written and has a clear rationale, the title and the abstract require attention. In the title, a verb is missing, while in the abstract, a simple comparison between the results of European studies and the meta-analysis is required so as not to confuse the reader.  The authors should clarify why many manuscripts (4195) were discarded and why manuscripts from 1957 were used for comparison due to the reduced antibiotic options. Also, it would be interesting to see in the analysis of the mono and combined therapy results in gram-negative vs. gram-positive infections, as is usually done in clinics. The analysis should be done despite the clear statistical constraints since it is clinically valuable. The authors should also analyze the limitations of antibiotic therapy based on the time of the initial treatment, organ involvement, and age of the studied group. Most manuscripts do not refer to age and comorbidities, but this issue should be discussed as the pharmacokinetics of the drugs can be affected. I would recommend the authors add a section on limitations to the study and a clear rationale of how studies should be designed to facilitate the conclusion of the effectiveness of the treatment.

Round 2

Reviewer 2 Report

Comments and Suggestions for Authors

Please change Coxiella brunetti to Coxiella burnetii.

Monotherapy should not be considered an empirical treatment (initial treatment) for infective endocarditis in all patients, according to the recent guidelines.

I understand the lack of information and the difficulty in assessing the true facts.